# 17β-Estradiol Promotes Tumorigenicity Through an Autocrine AREG/EGFR Loop in ER-α-Positive Breast Cancer Cells

**DOI:** 10.3390/cells14100703

**Published:** 2025-05-12

**Authors:** Sun Young Yoon, Yisun Jeong, Jai Min Ryu, Se Kyung Lee, Byung Joo Chae, Jonghan Yu, Seok Won Kim, Seok Jin Nam, Sangmin Kim, Jeong Eon Lee

**Affiliations:** 1Department of Breast Cancer Center, Samsung Medical Center, 81 Irwon-Ro, Gangnam-gu, Seoul 06351, Republic of Korea; soso66772000@naver.com (S.Y.Y.); sunrise1526@naver.com (Y.J.); jaimin.ryu@samsung.com (J.M.R.); sekyung.lee@samsung.com (S.K.L.); bj.chae@samsung.com (B.J.C.); jonghan.yu@samsung.com (J.Y.); seokwon1.kim@samsung.com (S.W.K.); seokjin.nam@samsung.com (S.J.N.); 2Department of Health Sciences and Technology, Samsung Advanced Institute for Health Sciences & Technology (SAIHST), Sungkyunkwan University, 81 Irwon-Ro, Gangnam-gu, Seoul 06355, Republic of Korea; 3Department of Surgery, Samsung Medical Center, Sungkyunkwan University School of Medicine, 81 Irwon-Ro, Gangnam-gu, Seoul 06351, Republic of Korea

**Keywords:** AREG, EGFR, ER, prognosis, targeted therapy

## Abstract

We previously reported that the level of EGFR expression is directly associated with the survival rate of estrogen receptor-positive (ER+) breast cancer patients. Here, we investigated how ER activation by 17β-estradiol (E2), the most potent form of estrogen, affects the expression or activity of EGFR or EGFR-related genes in ER+ breast cancer cells. As expected, E2 enhanced cell proliferation, the induction of S phase, and tumor growth in ER+ breast cancer models. E2 also increased the expression of secretory proteins, including amphiregulin (AREG), angiogenin, artemin, and CXCL16. We focused on AREG, which is a ligand of the epidermal growth factor receptor (EGFR). The levels of AREG expression were positively correlated with ESR1 expression. Our results also showed higher AREG mRNA expression levels in ER+ breast cancer cells than in ER- breast cancer cells. We treated ER+ breast cancer cells with lapatinib to inhibit the AREG/EGFR signaling pathway and then completely inhibited E2-induced cell proliferation and S-phase induction. Similar to the lapatinib treatment, cell proliferation, S-phase induction, cell migration, and tumor growth were suppressed by AREG knockdown. Taken together, we demonstrated that the induction of AREG by E2 contributes to EGFR activation, which then affects cell proliferation and tumor growth. Therefore, we suggest that AREG acts as an intermediary between EGFR and ER and targeting both ERs and EGFRs through combination therapy could prevent tumor progression in EGFR+ ER+ breast cancer patients.

## 1. Introduction

Breast cancer is the most commonly diagnosed cancer and the leading cause of cancer-related deaths in women worldwide [1]. Breast cancer is classified according to the presence or absence of estrogen receptor (ER), progesterone receptor (PR), and human epidermal growth factor receptor 2 (HER2) [2]. Nearly 70% of all breast cancers are ER+ and depend on estrogen to drive tumor growth [3]. ER-targeted endocrine therapy drugs, such as tamoxifen and fulvestrant, have been developed and are widely used to improve the survival of ER+ breast cancer patients [4,5]. However, despite exploring various therapeutic approaches, a high relapse rate remains a major clinical challenge.

In ER+ breast cancer, 17β-estradiol (E2) modulates the expression of specific genes that contribute to tumor progression through binding with ERs [6,7]. The E2/ER complex controls various cellular events, including cell proliferation, differentiation, and invasion [8]. E2 upregulates proangiogenic factors such as vascular endothelial growth factor (VEGF) and platelet-derived growth factor BB (PDGF-BB) and promotes resistance to antiangiogenic therapies by enhancing tumor vascular pericyte coverage and myeloid recruitment in non-small cell lung cancer [9,10]. We previously reported that epidermal growth factor receptor (EGFR)+ ER+ breast cancer was associated with a poor patient prognosis compared to patients with EGFR- and ER+ breast cancer [11]. However, the exact mechanism of the interaction between EGFR and ER on the prognosis of patients with ER+ breast cancer has not been elucidated. Therefore, the aim of this study was to find an intermediary between EGFR and ER.

Amphiregulin (AREG) is an EGFR ligand, along with EGF, transforming growth factor-α (TGFA), heparin-binding EGF-like growth factor (HBEGF), betacellulin (BTC), epiregulin (EPR), and epigen (EPGN) [12,13]. Elevated AREG expression has been observed in multiple cancer types, including breast, lung, and colorectal cancers, and promotes tumor progression, including cell proliferation, survival, and metastatic potential, by activating EGFR and its downstream signaling pathways such as MAPK, STAT3, and PI3K/AKT [14,15]. Xu et al. reported that stromal AREG induced programmed cell death 1 ligand (PD-L1) expression in prostate cancer cells and created an immunosuppressive tumor microenvironment (TME) against cytotoxic lymphocytes via immune checkpoint activation [16]. Therefore, we suggest that targeting AREG or the AREG-dependent signaling pathway could contribute to minimizing drug resistance and cancer cell proliferation.

The purpose of this study was to elucidate the role of AREG in promoting the aggressiveness of ER+ breast cancer by activating the E2/ER signaling pathway. In particular, we focused on understanding the mechanism of autocrine loops of AREG and why EGFR+ and ER+ breast cancer patients is associated with poor prognosis. Furthermore, suppressing the AREG/EGFR signaling pathway can be a fundamental therapeutic strategy for treating ER+ breast cancer when combined with classical chemotherapy.

## 2. Materials and Methods

### 2.1. Reagents

Human recombinant amphiregulin (AREG; cat. CYT-041), artemin (ARTN; cat. CYT-306), angiogenin (ANG; cat. PRO-1903), and CXCL16 (cat. CHM-029) were purchased from ProSpec-Tany TechnoGene Ltd. (Rehovot, Israel). 17β-estradiol (cat. S1709) and lapatinib (cat. S2111) were obtained from Selleck Chemicals (Houston, TX, USA).

### 2.2. Cell Culture Conditions

Human breast cancer cell lines were obtained from the American Type Culture Collection (ATCC), and their authenticity was confirmed through short tandem repeat profiling. Hs578T (ER-/PR-/HER2-), MDA-MB468 (ER-/PR-/HER2-), MDA-MB453 (ER-/PR-/HER2+), and MCF7 (ER+/PR+/HER2-), were cultured in Dulbecco’s Modified Eagle Medium (SH30243, HyClone, Cytiva), whereas T47D (ER+/PR+/HER2-), ZR75-1 (ER+/PR+/HER2-), BT474 (ER+/PR+/HER2+), HCC1143 (ER-/PR-/HER2-), and SKBR3(ER-/PR-/HER2+) cells were maintained in RPMI1640 medium (SH30027, HyClone, Cytiva). All culture media were supplemented with 10% fetal bovine serum (HyClone, Logan, UT, USA) and antibiotic-antimycotic solution (100X) (Gibco BRL Co., Gaithersburg, MD, USA) and incubated at 37 °C with 5% CO_2_. For E2 assays, cells were cultured in charcoal-stripped FBS (Gibco BRL Co., Gaithersburg, MD, USA) [17].

### 2.3. Generation of Stable AREG Knockdown Cells

Short hairpin RNA (shRNA) lentiviral vectors against human AREG (cat TL314703, OriGene, Rockville, MD, USA) or empty were transfected into 293 FT cells with pMD2G and psPAX2 envelope plasmids to generate stable AREG knockdown cell lines. The lentivirus was obtained after 2 days of transfection. T47D and MCF7 breast cancer cells were incubated with 2 mL of antibiotic-free medium containing 200 μL of lentivirus. Infected cells were selected using 1 µg/mL puromycin to generate stable cell lines. AREG expression levels in the generated cell lines were analyzed using real-time PCR.

### 2.4. Cell Proliferation Assay

T47D and MCF7 breast cancer cells were seeded in 96-well plates at a density of 2 × 10^3^ cells per well in 100 µL of medium and incubated at 37 °C. After 24 h, cells were treated with E2 at the indicated concentrations for 48 h. Cell proliferation was measured using a 3-(4,5-dimethylthiazol-2-yl)-2,5-diphenyltetrazolium bromide (MTT) assay by adding 10 µL of 5 mg/mL MTT solution (Sigma, St. Louis, MO, USA) per well, followed by optical density measurement at 590 nm using a microplate reader (SpectraMax 190, Molecular Devices, Sunnyvale, CA, USA). In parallel, cells were labeled with the CellTrace™ Far Red Cell Proliferation Kit (Life Technologies, Carlsbad, CA, USA) and analyzed after 72 h using a BD FACS verse flow cytometer (Becton-Dickinson, San Diego, CA, USA). Proliferation was evaluated by comparing far-red fluorescence intensity between control (0 h) and 72 h samples following treatment with E2 or AREG.

### 2.5. Cell Cycle Analysis

We performed flow cytometry with propidium iodide (PI)-stained cells to analyze the cell cycle. Trypsinized cells were fixed with 70% ethanol at room temperature (RT) for 1 h. After fixation, cells were treated with 100 µg/mL DNase-free RNase A (Thermo Fisher Scientific Biosciences GmbH, St. Leon-Rot, Germany) at 37 °C for 30 min and analyzed using a BD FACS Verse flow cytometer (Becton, Dickinson).

### 2.6. Colony Formation Assay

For the colony formation assays, T47D and MCF7 cells (1000 cells/well) were seeded into six-well plates and incubated overnight at 37 °C. Subsequently, each cell was treated with 100 nM E2, 50 ng/mL AREG, ANG, CXCL16, and ARTN, followed by an additional 10~14 days of incubation. The cells were fixed with 100% ethanol and stained with 0.04% crystal violet, and cell colonies were visualized using a CK40 inverted microscope (Olympus, Tokyo, Japan).

### 2.7. Wound-Healing Assay

T47D and MCF7 cells were seeded into six-well plates and cultured for 24 h. The cells were maintained in culture medium without FBS for 16–24 h. The cell monolayer was scratched with a 100 µL pipette tip to create a wound and washed twice with PBS to remove the suspended cells. The cells were maintained for 48 h in serum-containing medium, and cells migrating from the leading edge were photographed at 0 and 48 h using a CK40 inverted microscope (Olympus).

### 2.8. Cell Sphere Formation Assay

T47D and MCF7 cells (1000 cells/well) were seeded into six-well plates. Subsequently, each cell was treated with 100 nM E2, 50 ng/mL AREG, and/or 5 µM lapatinib. Each cell type was seeded in low adherent six-well plates and grown in serum-free DMEM/F12 media (SH30023, Hyclone, Cytiva) supplemented with B27 (1×, Invitrogen, Carlsbad, CA, USA), epidermal growth factor (EGF) at 20 ng/mL and fibroblast growth factor (FGF) at 20 ng/mL. Cell morphology was visualized using a CK40 inverted microscope (Olympus).

### 2.9. Quantitative Reverse Transcription PCR (RT-qPCR)

Total RNA was extracted from cultured cell lines using TRI Reagent (Molecular Research Center, Inc., Cincinnati, OH, USA) according to the manufacturer’s instructions. Complementary DNA (cDNA) was synthesized from 1 µg of RNA in a 20 µL reaction volume using the RevertAid First Strand cDNA Synthesis Kit (Thermo Fisher Scientific, Agawam, MA, USA). Gene expression was quantified by real-time PCR using the SensiMix SYBR Kit (Bioline Ltd., London, UK) on an ABI PRISM 7900HT instrument (Applied Biosystems, Foster City, CA, USA). Data were normalized to ACTB expression (F: TCA CCA ATT GGA TGA GCG GTT; R: AGT TTC GTG GAT GCC ACA GGA C) as a housekeeping gene, and relative expression levels were calculated using the 2^−ΔΔCT^ method. The specific primers for human AREG were F: AGG GTT GCC AGA TGC AAT AC and R: AAA CCA AGG CAC AGT GGA AC. Thermal cycling conditions were as follows: 50 °C for 2 min, 95 °C for 10 min, followed by 40 cycles of 95 °C for 15 s, 60 °C for 15 s, and 72 °C for 15 s.

### 2.10. Secretory Protein Array

T47D cells were seeded into two separate 100 mm dishes. Each cell plate was treated with DMSO or 100 nM E2 in fresh serum-free media for 24 h. Conditioned culture media were collected 24 h later. Next, 1 mL of conditioned culture media was used to analyze secretory proteins. A proteome Profiler™ Human Angiogenesis Array Kit (R&D Systems, Minneapolis, MN, USA) was used to detect the relative expression levels of 55 angiogenesis-related proteins according to the manufacturer’s instructions.

### 2.11. Enzyme-Linked Immunosorbent Assay (ELISA)

Secreted AREG and E2 protein levels in conditioned culture media (50 μL) and the serum (50 μL) of ER+ breast cancer patients were analyzed using the Human Amphiregulin ELISA Kit (DAR00, R&D Systems) and the Estradiol Parameter Assay Kit (KGE014, R&D Systems), according to the manufacturer’s protocol.

### 2.12. Correlation Analysis

Public AREG and ESR1 expression data in breast cancer cell lines were obtained from the Cancer Cell Line Encyclopedia (CCLE) database https://depmap.org/portal/ (accessed on 19 November 2024). The acquired data were analyzed and calculated using GraphPad Prism 8 software (GraphPad Software, La Jolla, CA, USA).

### 2.13. Approval of the Use of Human Breast Cancer Patient Serum Samples

The bio-specimens used in this study were provided by Samsung Medical Center BioBank (2023-0013) (IRB File No. SMC 2023-04-017). All pathological data, including ER status, PR status, HER2 status, pathological grading, lymph node metastasis status, and TNM stage, were obtained from the Samsung Medical Center BioBank. The acquired data were analyzed and calculated using GraphPad Prism 8 software (GraphPad Software, La Jolla, CA, USA).

### 2.14. Xenograft Study

Mice were housed in pathogen-free conditions in accordance with the Institute for Laboratory Animal Research Guide for the Care and Use of Laboratory Animals, under protocols approved by the Institutional Review Board of the Samsung Medical Center (Seoul, Republic of Korea). Six- to eight-week-old female Balb/c nude mice (18–22 g; Orient Bio, Seoul, Republic of Korea) were used to establish a xenograft model. For mammary fat pad injections, 5–7-week-old mice were anesthetized and implanted with estrogen pellets. After 24 h, tumor cells were injected into the second mammary fat pad. Mice were weighed and tumors were manually palpated and measured using a digital caliper three times per week until euthanasia. Tumor size was calculated using the formula: tumor volume = (width^2^ × length)/2 (mm^3^). Throughout the experiment, animals were monitored for morbidity, weight loss, and tumor ulceration. Tumors were excised and subjected to histological analysis by hematoxylin and eosin (H&E) staining, as well as Ki67 and AREG immunohistochemistry.

### 2.15. Immunohistochemical Staining

Xenograft tissues were formalin-fixed, paraffin-embedded, sectioned at 4 μm thickness, and processed through deparaffinization in xylene, graded alcohol dehydration, and rehydration in water. Sections were stained with hematoxylin and eosin (H&E) for histological evaluation, and immunohistochemistry (IHC) was performed to detect Ki-67 (Abcam, Cambridge, UK) and amphiregulin (AREG; Proteintech Group, Rosemont, IL, USA) using appropriate positive and negative controls at the Animal Pathology Core Laboratory of the Samsung Medical Center. After washing, sections were incubated with a biotinylated goat anti-mouse or anti-rabbit secondary antibody (Dako, Campbellfield, Australia), followed by streptavidin–horseradish peroxidase (HRP) complex incubation. Immunoreactivity was visualized with 3,3′-diaminobenzidine (DAB) chromogen (Dako Liquid DAB Plus, K3468) developed for 5 min. Sections were subsequently treated with streptavidin (BD Pharmingen, CA, USA) and developed with DAB tetrahydrochloride (BD Pharmingen). Quantitative Ki-67 expression was assessed by counting positive cells in four random fields per slide. Images were analyzed using a Scanscope XT system (Aperio Technologies, CA, USA).

### 2.16. Statistical Analysis

Data analysis and graph generation were performed using Microsoft Excel 2016 (Microsoft, Redmond, WA, USA) and GraphPad Prism 8 (GraphPad Software, La Jolla, CA, USA). Data are presented as the mean ± standard error of the mean (SEM). Statistical significance was evaluated using one-way analysis of variance (ANOVA) or an unpaired, two-tailed Student’s *t*-test. A *p*-value < 0.05 was considered statistically significant. All experiments were performed independently in triplicate or more.

## 3. Results

### 3.1. E2 Increases Cell Proliferation, Growth, and Tumorigenicity in ER+ Breast Cancer Cells

Most of the actions of estrogens directly control the cell cycle by activating ERs and upregulating proteins such as cyclin D1 and A [18]. These effects were verified by confirming the effect of E2 on the cell proliferation, growth, and tumorigenicity of ER+ breast cancer cells [8]. Here, we utilized charcoal serum stripping to study the effects of E2 in vitro without the confounding effects of hormones endogenous to fetal bovine serum (FBS). The E2 treatment increased cell proliferation in a dose-dependent manner (Figure 1A). We selected 100 nM E2 to perform the subsequent experiments. We also observed that E2 stimulated S-phase entry in T47D and MCF7 breast cancer cells (Figure 1B). Twenty-four hour treatment of T47D and MCF7 breast cancer cells with E2 increased the S-phase ratio by 8.5 ± 1.3% and 4.5 ± 1.2%, respectively. E2 treatment also increased cell growth, as shown by the colony-forming assay (Figure 1C). Similar to these results, the number and size of spheres were significantly increased by E2 in T47D and MCF7 breast cancer cells (Figure 1D). An E2 pellet was subcutaneously inoculated into the mice, and T47D cells were injected into the second fat pad to confirm the effect of E2 in an in vivo mouse model. Four weeks later, the tumorigenicity of the T47D cells was increased by 8.2 ± 4.0-fold in the presence of E2 pellets compared to in the absence of E2 pellets (Figure 1E).

### 3.2. E2-Induced AREG Expression Stimulates Cell Proliferation

We analyzed secretory proteins following E2 treatment using a Proteome Profiler Human Array Kit. We treated T47D breast cancer cells with 100 nM E2 for 24 h under serum-free media conditions to identify the intermediary between ER and EGFR. The levels of angiogenin (ANG), CXCL16, artemin (ARTN), and AREG protein expression were significantly increased by E2 treatment (Figure 2A). However, recombinant human ANG, CXCL16, and ARTN proteins failed to greatly affect the cell cycle or growth (Appendix A). However, AREG acts as a significant downstream effector of ER signaling and is directly involved in cell proliferation [19]. Therefore, we assumed that AREG is an intermediary connecting ER and EGFR. So, we checked the levels of AREG mRNA and protein expression under the same conditions as those in Figure 2A. Basal levels of AREG mRNA (Figure 2B) and protein (Figure 2C) expression were significantly increased by E2 treatment. Like E2 treatment, AREG also increased S-phase entry (Figure 2D). AREG treatment for 24 h increased S-phase entry in T47D and MCF7 breast cancer cells by 4.9 ± 0.2% and 7.4 ± 0.2%, respectively (Figure 2D). We confirmed that AREG promoted cell proliferation (Figure 2E) and cell growth (Figure 2F) in both T47D and MCF7 cells through the fluorescence intensity of red dye and the colony-forming assay, respectively. In addition, AREG knockdown cell lines did not significantly affect cell proliferation (Figure 2G) and cell cycle progression (Figure 2H,I) in response to E2 treatment. Based on these results, we hypothesized that E2-induced AREG expression mediated the cell proliferation, growth, and tumorigenicity of ER+ breast cancer cells by activating the EGFR.

### 3.3. Comparison of Charcoal-Stripped Fetal Bovine Serum and Normal Fetal Bovine Serum on AREG Expression

Because charcoal-stripped fetal bovine serum (CS-FBS) is commonly utilized in research on hormone-sensitive cancers to establish hormone-depleted cell culture environments [20], we compared the effects of CS-FBS and normal FBS on AREG expression. Each cell type was seeded into six-well plates for 24 h, washed with phosphate-buffered saline (PBS), and then incubated with CS-FBS-containing media or FBS-containing media for 24 h. As shown in Figure 3A, the level of AREG mRNA expression was significantly increased in FBS-containing media. Secreted AREG protein expression was also increased under the same conditions (Figure 3B). Therefore, we checked the levels of estradiol in each medium using an ELISA kit. The level of estradiol was increased by 5.6 ± 2.0-fold in FBS-containing media compared to CS-FBS-containing media (Figure 3C). Next, we investigated cell cycle alterations under the same conditions. Our results showed that CS-FBS-containing media induced cell cycle arrest at the G0/G1 phase in both T47D and MCF7 cells (Figure 3D). In contrast, the S phase and the G2/M phase ratio were increased by FBS-containing media (Figure 3D). Next, we examined the induction of AREG by E2 treatment using xenograft models (Figure 3E,F). Immunohistochemical (IHC) analysis showed the increased expression of the proliferation marker Ki67 and AREG in the xenograft model with E2 pellets compared to the absence of E2 pellets (Figure 3E). Blood AREG levels were also elevated in the xenograft model with E2 pellets (Figure 3F).

### 3.4. Levels of E2 Are Positively Correlated with AREG Expression in ER+ Breast Cancer Patient Serum

We collected blood samples from ER+- breast cancer patients to verify the correlation between E2 and AREG (Table 1). We analyzed the levels of AREG and E2 expression in 100 blood samples using ELISA kits. As shown in Figure 4A, secreted AREG expression in serum was meaningfully increased in breast cancer patients with high E2. We also found that endogenous E2 expression was positively correlated with AREG expression (Figure 4B). Furthermore, we compared the expression of AREG and E2 according to pre- or post-menopausal status. The E2 levels in pre-menopausal patients were significantly higher than those in post-menopausal patients (Figure 4C). Consistent with this result, the levels of AREG expression were elevated in pre-menopausal patients with high E2 levels (Figure 4C).

Next, we analyzed the correlation between AREG and ESR1 using the published Cancer Cell Line Encyclopedia (CCLE) database. We found that ESR1 expression was also positively correlated with AREG expression in a variety of breast cancer cell lines (Figure 4D). Additionally, we found that the levels of AREG expression were higher in ER+ breast cancer patients than in ER- breast cancer patients using the E-TABM-158 dataset (Appendix A). The levels of AREG mRNA and protein expression were increased in ER+ breast cancer cell lines compared to ER- breast cancer cell lines (Appendix A). These results suggest that E2 expression levels play an important role in AREG expression. E2, as well as ESR1 expression, was positively correlated with AREG expression in breast cancer models.

### 3.5. E2 Activates the EGFR Signaling Pathway via an Autocrine AREG Loop

ER+ breast cancer cells were incubated with or without E2-containing media for 24 h to investigate the effect of E2-induced AREG on the EGFR signaling pathway. Twenty-four hours later, we harvested conditioned culture media (CM) and added new plates with ER+ breast cancer cells, with or without lapatinib, for 24 h. A schematic diagram of the experimental process is illustrated in Figure 5A. After sample collection, we analyzed the expression of AREG mRNA. As shown in Figure 5B, the level of AREG mRNA expression in both T47D and MCF7 breast cancer cells was increased by E2 CM treatment. Our results showed that E2 CM-induced AREG expression was not changed by lapatinib treatment (Figure 5B). However, under the same conditions, E2 CM-induced cell growth, including cell size and numbers, was completely blocked by lapatinib treatment (Figure 5C). Compared to CON CM treatment, cell cycle S-phase entry was triggered by E2 CM (Figure 5D). In contrast, S-phase entry by E2 CM was shifted to G0/G1 arrest by lapatinib (Figure 5D). In parallel, empty vector control or AREG knockdown cell lines were incubated with or without E2 for 24 h, and the CM were collected. Treatment with the four different CM showed that AREG knockdown diminished the effects of E2, resulting in no significant induction of S-phase entry or sphere growth (Appendix A). These results suggest that the induction of AREG by E2 mediates cell growth by activating the EGFR signaling pathway.

### 3.6. AREG Knockdown Suppresses S-Phase Entry and Cell Migration in ER+ Breast Cancer Cells

We silenced AREG expression in T47D and MCF7 cells using a lentiviral system to demonstrate its direct effect on the cell cycle or cell migration. The levels of AREG mRNA and protein expression were decreased by AREG knockdown (Figure 6A,B). Next, we analyzed cell migration. The cell-free area was decreased in ER+ breast cancer cells treated with vector only (Figure 6C). However, when AREG was knocked down, the cell-free area did not change significantly (Figure 6C). AREG knockdown decreased the migration distance of T47D and MCF7 breast cancer cells to 18.1 ± 3.8 pixels and 27.1 ± 2.5 pixels, respectively (Figure 6C). The increase in S-phase entry induced by AREG or E2 treatment was significantly suppressed upon AREG knockdown (Figure 6D). AREG knockdown decreased S-phase cell cycle entry to 4.9 ± 0.1% (in T47D cells) and 4.5 ± 0.6% (in MCF7 cells) (Figure 6D). The reduction in cell migration and the S-phase entry by AREG knockdown were rescued by treatment with recombinant AREG (Figure 6C,D). Thus, we suggest that AREG directly regulates cell proliferation and migration in ER+ breast cancer cells.

### 3.7. Diminished Expression of AREG Inhibits Tumorigenicity in an Orthotopic Xenograft Model

Lastly, we evaluated the effect of AREG knockdown on the tumorigenicity of T47D breast cancer cells using an orthotopic xenograft model. T47D cells with vector alone (shCON) or with AREG knockdown (shAREG) were injected into the second fat pad. After 60 days, we harvested tumor tissues. A schematic diagram of the experimental process is illustrated in Figure 7A. As expected, tumor volume was significantly decreased in the shAREG group (Figure 7B). IHC staining was used to analyze the levels of Ki67 (cell proliferation marker) and AREG. The expression of the proliferation marker Ki67 was decreased in AREG knockdown tumors (Figure 7C). We found that AREG expression in the serum of tamoxifen-treated ER+ breast cancer patients increased in a stage-dependent manner, although there were no stage 4 breast cancer patients in our sample (Figure 7D). Also, similar to the results in Figure 7C, AREG expression was increased in ER+ breast cancer patients with high Ki67 levels compared to those with low Ki76 levels (Figure 7E). Thus, the in vitro or in vivo data suggest that AREG plays an important role in the tumorigenicity of ER+ breast cancer.

## 4. Discussion

Estrogens are steroid hormones that can be classified into three major types of biologically active estrogens in women: estrone (E1), E2, and estriol (E3) [21]. During life, the predominant female sex hormone is estradiol (E2), estriol (E3) is the main hormone during pregnancy, and estrone (E1) is the main hormone in the post-menopausal period [22]. E2 not only regulates multiple physiological processes to maintain homeostasis in normal conditions, but it is also responsible for the development of malignant tumors, including breast, prostate, gynecologic, lung, and colorectal cancer [23]. Consistent with these results, our results showed that E2 enhanced the cell proliferation, migration, and tumorigenicity of ER+ breast cancer cells. In general, CS-FBS nearly completely removed hormonal factors (e.g., estrogen, androgen, and progesterone) [24]. As expected, the level of E2 in normal FBS was more than four times higher than in CS-FBS. The growth and S-phase entry of ER+ breast cancer cells cultured in normal FBS were superior to those cultured in CS-FBS. These phenomena were completely inhibited by fulvestrant treatment [19].

E2 can affect multiple pathways in a variety of cells by encoding cytokines and factors associated with the immune response, signal transduction, cell migration, and cytoskeleton regulation [25]. Consistent with these results, our results showed that E2 significantly increased the levels of oncogenic secretory proteins, such as AREG, ANG, CXCL16, and ARTN, in ER+ breast cancer cells. Remarkably, other secretory proteins, such as ANG, CXCL16, and ARTN, had no effect on the growth of ER+ breast cancer. Only AREG affected cell proliferation, S phase entry, and migration. AREG, an EGFR ligand, plays a central role in mammary gland development and branching morphogenesis control in various organs [26]. AREG is expressed in tumor cells, including breast, colon, and lung, but not in stromal or inflammatory cells and has been implicated in the initiation and progression of tumors [27]. However, the mechanism of the interaction between ER and EGFR in ER+ breast cancer has not been fully elucidated. Therefore, this study sought to identify the important mediators in the relationship between ER and EGFR and focused on the study of their function.

Serum levels of AREG were higher in 41 patients with colorectal cancer (stage IV or recurrent group, median = 31.55 pg/mL) than in the stage I–III group (78 patients, median = 15.48 pg/mL) [28]. Consistent with these results, we also found that secreted AREG protein in serum increased in a stage-dependent manner, although there are no stage IV breast cancer patients in our sample of tamoxifen-treated ER+ breast cancer patients. Peterson et al. reported that serum AREG levels no difference in women with active breast cancer (43 with ER+ and 34 with ER-) or between this cohort and the normal population [29]. However, our results revealed that serum levels of AREG were higher in pre-menopausal breast cancer patients than in post-menopausal patients. In addition, the levels of E2 were increased in pre-menopausal breast cancer patients. The levels of AREG expression were positively associated with E2 as well as ESR1 expression. These results demonstrated that high serum AREG is associated with advanced ER+ breast cancers and levels of E2 and ESR1 expression.

E2-induced mitogen-activated protein kinase (ERK1/2) activation requires the expression of the G protein-coupled receptor homolog, GPR30 [30,31]. In addition, E2 signaling to ERK 1/2 was dependent upon the transactivation of the EGFR via the release of membrane-associated heparin-bound EGF (HB-EGF) in MDA-MB231 cells, which are ER- breast cancer cells [30]. However, E2 treatment of T47D ER+ breast cancer cells did not increase HB-EGF (Figure 2A). Based on these results, we think that AREG may be an intermediary between E2 and EGFR. In the absence of AREG expression, cell growth was slowed, and the invasion of inflammatory cells into breast cancer cells was reduced but not completely inhibited [32]. Tumor formation was also decreased in xenograft models of ER+ breast cancer cells [19]. Consistent with these reports, we also found that cell proliferation, as well as migration, was significantly decreased by AREG knockdown in ER+ breast cancer cells. Therefore, the induction of AREG by E2 plays an important role in cell proliferation and migration in EGFR+ and ER+ breast cancer cells.

## 5. Conclusions

In conclusion, in this study, we focused on understanding the mechanism of autocrine loops of AREG and why EGFR+ and ER+ breast cancer is associated with poor patient prognosis. The first approach to unlocking this mechanism analyzed the secretory proteins induced by E2. We found that E2 increased various proteins such as AREG, ANG, CXCL16, and ARTN. Among these, AREG was the only protein that promoted the growth of ER+ breast cancer cells. The cell proliferation, migration, and tumorigenicity of ER+ breast cancer cells were increased by recombinant human AREG treatment. In contrast, these phenomena were suppressed by AREG knockdown. E2-induced cell proliferation and migration were also inhibited by lapatinib, a specific EGFR inhibitor. Therefore, our findings suggest that AREG plays an important role as an intermediary between ER and EGFR. Furthermore, suppressing the AREG/EGFR signaling pathway can be a fundamental therapeutic strategy for EGFR+ and ER+ breast cancers when combined with classic chemotherapy.

## Figures and Tables

**Figure 1 cells-14-00703-f001:**
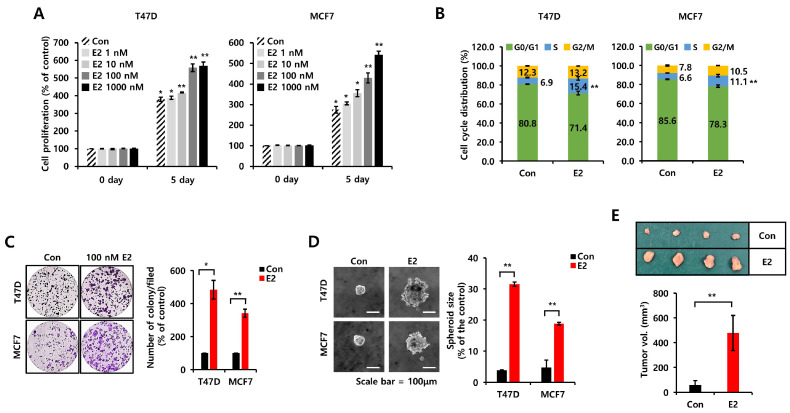
E2 increases cell proliferation, growth, and tumorigenicity in ER+ breast cancer cells. (**A**) Cell proliferation was analyzed in T47D and MCF7 cells treated with or without the indicated concentration of E2 for 5 days using the MTT assay. (**B**) The cell cycle of T47D and MCF7 cells treated with or without 100 nM E2 for 24 h was analyzed by flow cytometry. (**C**) Cell growth was analyzed by colony-forming assays. Each cell type was seeded in a six-well plate for 24 h and then treated with 100 nM E2 for 14 days. (**D**) Each cell type was seeded in low-adherent six-well plates in serum-free culture media with growth factors and treated with or without 100 nM E2 for spheroid formation. Scale bar, 100 µm. Values are shown as the mean ± SEM. All experiments were performed in triplicate. (**E**) The tumorigenicity of T47D cells in xenograft mouse models with implanted estrogen pellets. Estrogen pellets were subcutaneously implanted, and T47D cells were injected into the second mammary fat pad after 24 h. Tumor tissues were harvested from the mice for tumor volume measurements and tissue staining. All *p*-values were calculated using unpaired two-tailed Student’s *t*-tests. * *p* < 0.05, ** *p* < 0.01.

**Figure 2 cells-14-00703-f002:**
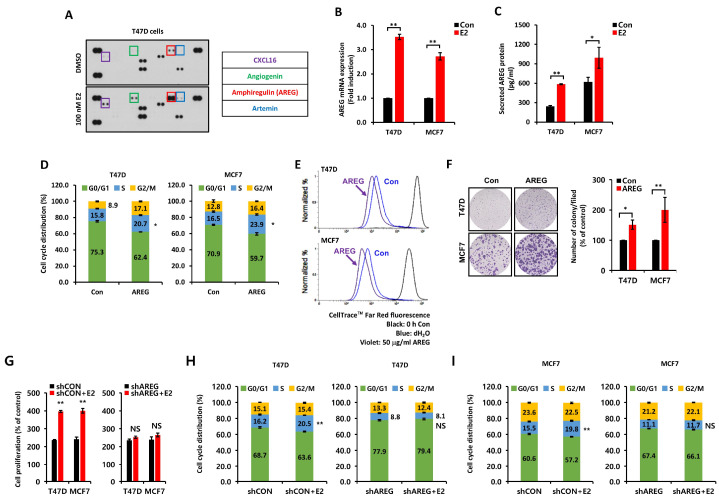
E2-induced AREG expression stimulates cell proliferation. (**A**) Secreted protein expression was analyzed using the proteome Profiler™ Human Angiogenesis Array Kit. T47D cells were treated with or without 100 nM E2 for 48 h. Conditioned culture media were harvested for secreted protein analysis. (**B**,**C**) The levels of AREG mRNA and protein were analyzed by real-time PCR (**B**) or ELISA (**C**). Each cell type was treated with or without 100 nM E2 for 24 h. (**D**) Cell cycles in T47D and MCF7 cells treated with or without 50 ng/mL AREG for 24 h were analyzed by flow cytometry. (**E**) Cell proliferation changes induced by AREG treatment were analyzed using CellTrace^TM^ Far Red fluorescence. (**F**) Cell growth was analyzed using colony-forming assays. Each cell type was seeded in a six-well plate for 24 h and then treated with 50 ng/mL AREG for 14 days. (**G**) shCON or shAREG-transfected cells were incubated with or without E2 for 3 days, and cell proliferation was measured using the MTT assay. (**H**,**I**) T47D and MCF7 cells transfected with shCON or shAREG were incubated with or without E2 for 24 h, followed by cell cycle analysis using flow cytometry. All experiments were performed in triplicate. All *p*-values were calculated using unpaired two-tailed Student’s *t*-tests. NS: non significant * *p* < 0.05, ** *p* < 0.01.

**Figure 3 cells-14-00703-f003:**
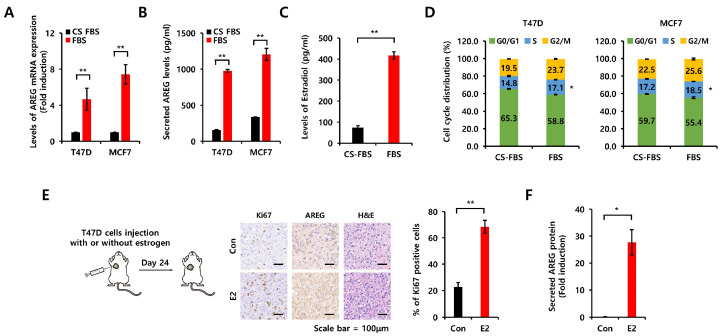
Comparison of charcoal-stripped fetal bovine serum and normal fetal bovine serum on AREG expression. (**A**,**B**) Each cell type was seeded into six-well plates under CS-FBS and normal FBS conditions for 48 h. After 48 h, cell lysates and conditioned culture media were harvested for mRNA analysis by real-time PCR (**A**) and proteins by ELISA (**B**). (**C**) Endogenous E2 levels were analyzed by ELISA in culture media with CS-FBS or FBS. (**D**) The cell cycle was analyzed by flow cytometry under CS-FBS and normal FBS conditions. All experiments were performed in triplicate. (**E**) After estrogen pellet implantation, T47D cells were injected into the second mammary fat pad of BALB/c mice, and tumor growth was monitored. The expression levels of AREG and Ki67 in the xenografts were analyzed by immunohistochemical staining. Quantitative data for Ki67-positive cells were obtained by counting four fields. Con group (*n* = 4), E2 group (*n* = 4). Values are the mean ± SEM. Scale bar, 100 µm. (**F**) Secreted AREG protein levels in mouse serum samples were qualified by ELISA analysis, comparing the estrogen pellet implantation group to the control group. * All *p*-values were calculated using unpaired two-tailed Student’s *t*-tests. * *p* < 0.05, ** *p* < 0.01.

**Figure 4 cells-14-00703-f004:**
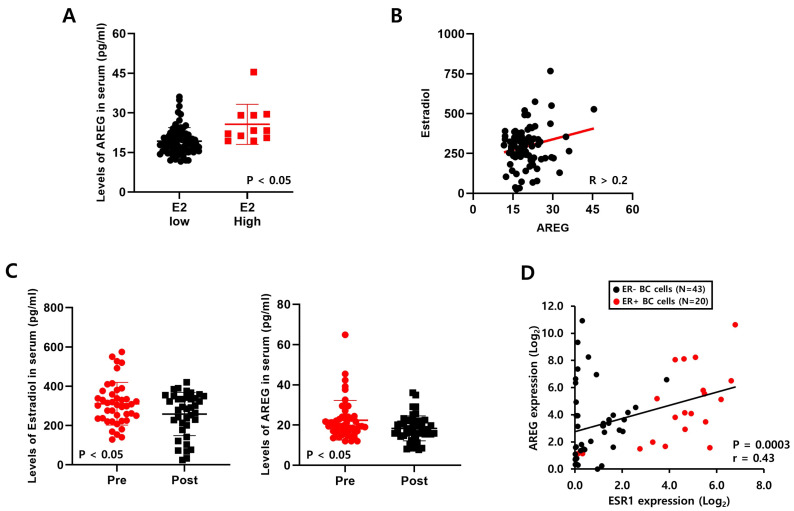
Levels of E2 are positively correlated with AREG expression in ER+ breast cancer patient serum. (**A**) The levels of AREG protein in serum samples from ER-positive breast cancer patients were analyzed by ELISA. (**B**) The correlation between estradiol and AREG secretion levels in ER-positive breast cancer patient serum samples (*n* = 100). (**C**) AREG protein expression and estrogen levels were analyzed according to menopausal status. (**D**) The correlation between ESR1 and AREG expression levels was analyzed using the CCLE database. Values are the mean ± SEM. All *p*-values were calculated using one-way analysis of variance (ANOVA) or the Student’s *t*-test (unpaired, two-tailed).

**Figure 5 cells-14-00703-f005:**
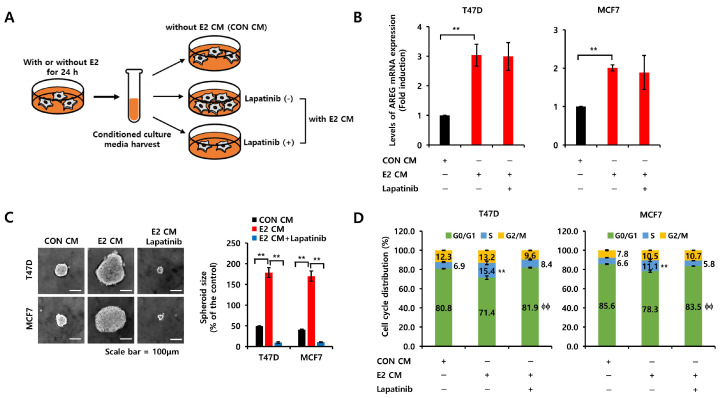
E2 activates the EGFR signaling pathway via an autocrine AREG loop. (**A**) Schematic model of the experimental process. (**B**) Each cell was treated with CON CM, E2 CM, or E2 CM with lapatinib for 24 h. The levels of AREG mRNA expression were analyzed by real-time PCR. (**C**) Each cell type was seeded in low-adherent six-well plates containing CON CM, E2 CM, or E2 CM with lapatinib for spheroid formation. Scale bar, 100 µm. (**D**) The cells were treated with CON CM, E2 CM or E2 CM with lapatinib for 24 h. The cell cycle was analyzed by flow cytometry. All experiments were performed in triplicate. Values are the mean ± SEM. All *p*-values were calculated using one-way analysis of variance (ANOVA) or the Student’s *t*-test (unpaired, two-tailed). ** *p* < 0.01, ϕϕ *p* < 0.01.

**Figure 6 cells-14-00703-f006:**
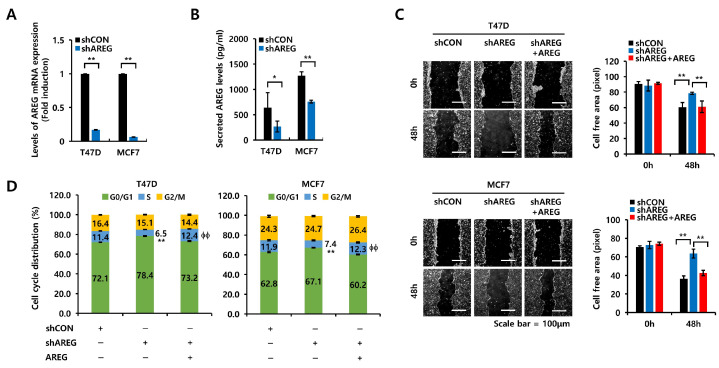
AREG knockdown suppresses S-phase entry and cell migration in ER+ breast cancer cells. (**A**,**B**) Each cell type was transfected with an empty vector (shCON) or shAREG lentivirus. Transfected cells were selected using puromycin. The levels of AREG mRNA and protein expression were analyzed by real-time PCR (**A**) and ELISA (**B**). (**C**) Cell migration was analyzed using the wound-healing assay. Scale bar, 100 µm. (**D**) Cell cycles in shCON or shAREG-transfected cells treated with or without recombinant AREG were analyzed by flow cytometry. All experiments were performed in triplicate. Values are the mean ± SEM. All *p*-values were calculated by unpaired two-tailed Student’s *t*-tests. * *p* < 0.05, ** *p* < 0.01, ϕϕ *p* < 0.01.

**Figure 7 cells-14-00703-f007:**
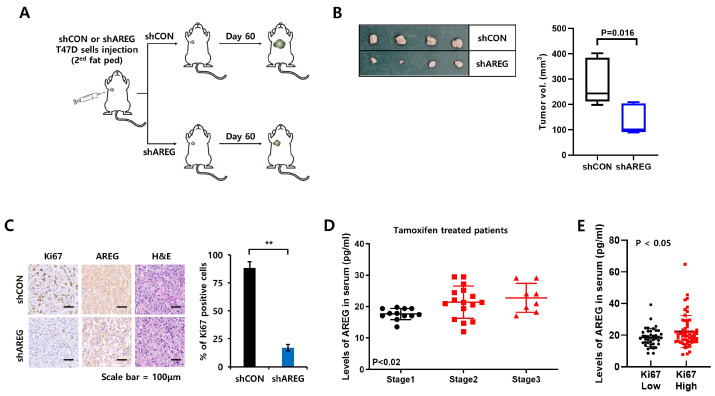
Diminished expression of AREG inhibits tumorigenicity in an orthotopic xenograft model. (**A**) Schematic model of the experimental process. (**B**) T47D cells with shCON or shAREG cells (1.8 × 10^7^ cells) were injected into the 2nd mammary fat pad of BALB/c mice, and tumor growth was monitored. shCON group (*n* = 4), shAREG group (*n* = 4). After 60 days, tumor tissues were harvested to measure tumor volume. (**C**) The expression levels of AREG and Ki67 in xenografts were analyzed by immunohistochemical staining. Quantitative data for Ki67-positive cells were obtained by counting four fields. Scale bar, 100 µm. (**D**,**E**) Human serum AREG levels in ER+ breast cancer patients were analyzed by ELISA according to TNM stage (**D**) or Ki67 status (**E**). Values are the mean ± SEM. All *p*-values were calculated using one-way analysis of variance (ANOVA) or the Student’s *t*-test (unpaired, two-tailed). ** *p* < 0.01.

**Table 1 cells-14-00703-t001:** Clinicopathological characteristics of serum samples from ER+ breast cancer patients.

ER+ Breast Cancer Patients		
Characteristics		*n* = 100
Age	Pre	41.76 (28–54)
	Post	57.02 (40–72)
T stage	T1	36
	T2	45
	T3	17
	T4	2
N stage	N0	39
	N1	29
	N2	18
	N3	14
M stage	M0	96
	M1	4
EGFR	−	88
	+	12
PR	−	21
	+	78
HER2	−	74
	+	26

## Data Availability

Data are contained within the article.

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
