# Peer review of "17β-Estradiol Promotes Tumorigenicity Through an Autocrine AREG/EGFR Loop in ER-α-Positive Breast Cancer Cells"

_cells, 2025, doi:10.3390/cells14100703_

Round 1
Reviewer 1 Report
Comments and Suggestions for Authors
Major comments:
- Figure 1 serves to inform E2 treatment boosts T47D/MCF7 fitness, but this is already known. It has also been already shown that AREG is responsive to E2 treatment in ER+ breast cancer cells, ex: PMID: 26527289 and acts on EGFR to regulate cell fitness, ex PMID: 18381432. There is no discussion to this effect in the text. The novelty of the study is therefore unclear.
- The authors need to provide more information in the methods section on the E2 used (aka vendor, catalog number), and how they obtained recombinant AREG (and other proteins), and what concentrations of these were used in their assays.
- Figure 2 shows recombinant AREG can boost cell fitness phenotypes in MCF7 and T47D cells. This suggests AREG is sufficient, but not that it is necessary. A number of other factors downstream of ER signaling might do the same. The secretory protein array covers 55 angiogenesis-related proteins and the authors show that the other ‘hits’ in this assay do not confer the same fitness, but this might not be true for other ER-related secreted proteins. The necessity experiment would come with E2 treatment in cells that have genetically silenced AREG.
- The experiments in Figure 5 do not get at the question of AREG-dependency and EGFR specificity. Increased AREG expression due to E2 treatment is used to infer its role in driving the observed response to lapatinib – without any modulation to AREG itself. Again, it could be any number of factors downstream of ER modulating this phenotype. Doing this experiment in a context where AREG is deleted (with and without rescue with recombinant protein) would lend confidence to a role it might play in a potential crosstalk between ER and EGFR. Additionally, it is unclear what concentration of lapatinib is used in this experiment – and as such, whether the fitness defects observed are on target.
- The experiments in Figure 6 showing knockdown of AREG are problematic because it is unclear whether the observed phenotypes are due to on-target effects or off-target toxicity which can be answered with a rescue experiment – this also applies to the xenograft model in Figure 7.
Minor comments:
- Font size for some figure panels make it hard to read them. These should be made larger for legibility. Examples include Figure 1B, 2A, 2D (and other cell cycle assay plots)
- Abstract – line 21 – ‘positive correlated with E2 expression’ – the authors likely mean ER here.
- Line 95 – Typo in 2 x 10^3 cells.
- Line 413 – ‘Authors’, presumably an error.
Author Response
Major comments:
- Figure 1 serves to inform E2 treatment boosts T47D/MCF7 fitness, but this is already known. It has also been already shown that AREG is responsive to E2 treatment in ER+ breast cancer cells, ex: PMID: 26527289 and acts on EGFR to regulate cell fitness, ex PMID: 18381432. There is no discussion to this effect in the text. The novelty of the study is therefore unclear.
Thank you for your comment. These research papers (PMID: 26527289 and PMID: 18381432) helped us a lot in setting our research direction. Also, we have been reported that the survival rates of EGFR+ER+ breast cancer patients are lower than that of EGFR-ER+ breast cancer patients (PMID: 31670920). However, the cause of low survival rates of EGFR+ER+ breast cancer patients was not well known. So, in this study, we would like to find an intermediary to connect the relationship between EGFR and ER.
- The authors need to provide more information in the methods section on the E2 used (aka vendor, catalog number), and how they obtained recombinant AREG (and other proteins), and what concentrations of these were used in their assays.
Thank you for your comment. We described in the material section as follows. “Human recombinant proteins including amphiregulin (Cat. CYT-041), artemin (Cat. CYT-306), angiogenin (Cat. PRO-1903), and CXCL16 (Cat. CHM-029) were pur-chased from ProSpec-Tany TechnoGene Ltd. (Rehovot, Israel). 17β-estradiol (Cat. S1709) and lapatinib (Cat. S2111) were obtained from Selleck Chemicals (Houston, TX, USA).”
Although data not shown, we tested the efficacies of recombinant in a dose-dependent manner (10, 25, 50 ng/ml concentration, respectively) in ER+ breast cancer cells. However, our results did not show statistically significant differences among concentrations. So, we selected 50 ng/ml concentration of each recombinant protein
- Figure 2 shows recombinant AREG can boost cell fitness phenotypes in MCF7 and T47D cells. This suggests AREG is sufficient, but not that it is necessary. A number of other factors downstream of ER signaling might do the same. The secretory protein array covers 55 angiogenesis-related proteins and the authors show that the other ‘hits’ in this assay do not confer the same fitness, but this might not be true for other ER-related secreted proteins. The necessity experiment would come with E2 treatment in cells that have genetically silenced AREG.
Thank you very much for your good question. We investigated the effect of E2 on AREG knockdown cell lines and added new data in supplement 1. As expected, AREG knockdown cell lines did not significantly affect cell cycle or cell proliferation by E2 treatment. So, we suggest that E2-induced AREG expression plays an important role on E2-induced cell cycle and cell proliferation.
- The experiments in Figure 5 do not get at the question of AREG-dependency and EGFR specificity. Increased AREG expression due to E2 treatment is used to infer its role in driving the observed response to lapatinib – without any modulation to AREG itself. Again, it could be any number of factors downstream of ER modulating this phenotype. Doing this experiment in a context where AREG is deleted (with and without rescue with recombinant protein) would lend confidence to a role it might play in a potential crosstalk between ER and EGFR. Additionally, it is unclear what concentration of lapatinib is used in this experiment – and as such, whether the fitness defects observed are on target.
Thank you very much for your good question. According to your comment, empty vector alone or AREG knockdown cell lines were incubated with or without E2 for 24 h and then harvested conditioned culture media. Under the four conditioned media, each cell was incubated for 48 h. As expected, upon AREG deletion, E2 CM did not significantly induce S phase entry, and sphere growth. However, in empty vector alone cell lines, E2 CM was dramatically increased S phase entry, and sphere growth. These findings suggest that AREG may function as a potential mediator on the relationship between ER and EGFR. We added new data in supplement 3.
Although data not showed, we examined the efficacies of lapatinib in a dose-dependent manner (0, 1, 5, 10 mM, respectively). Each lapatinib concentration did not induce significant cytotoxicity and so, choose 5 mM lapatinib concentration.
- The experiments in Figure 6 showing knockdown of AREG are problematic because it is unclear whether the observed phenotypes are due to on-target effects or off-target toxicity which can be answered with a rescue experiment – this also applies to the xenograft model in Figure 7.
Thank you for your comment. According to your comment, we performed a rescue experiment and added new data in Fig. 6C and 6D. As expected, AREG deletion reduced cell migration, which was subsequently rescued by recombinant AREG treatment. In addition, recombinant AREG restored the reduced S-phase entry caused by AREG knockdown.
Minor comments:
- Font size for some figure panels make it hard to read them. These should be made larger for legibility. Examples include Figure 1B, 2A, 2D (and other cell cycle assay plots).
Thank you for your comment. We have revised your comment. Specifically, we increased the size of the figure panels and fonts to improve legibility, and adjusted the resolution of all figures to 300 DPI. Figures 1B, 2A, 2D, and other cell cycle assay plots have been updated accordingly.
- Abstract – line 21 – ‘positive correlated with E2 expression’ – the authors likely mean ER here.
Thank you for your comment. It’s my mistake. I’m so sorry. I corrected. The levels of AREG expression were positively correlated with ESR1 expression.
- Line 95 – Typo in 2 x 10^3 cells.
Thank you for your comment. We have revised your comment.
Line 413 – ‘Authors’, presumably an error.
Thank you for your comment. It’s my mistake. I’m so sorry. I corrected.

Reviewer 2 Report
Comments and Suggestions for Authors
The manuscript entitled “17β-estradiol promotes cell proliferation through an autocrine AREG/EGFR loop in ER-α-positive breast cancer cells” by Yoon, S. Y. et al. investigates the effects of estrogen receptor (ER) activation by 17β-estradiol (E2), the most potent physiological estrogen, on the expression and activity of EGFR and EGFR-related genes in ER-positive breast cancer cells. The authors demonstrate that E2 enhances cell proliferation and tumor growth in ER-positive breast cancer models. Specifically, E2 upregulates the expression of the secretory protein amphiregulin (AREG), a ligand for the epidermal growth factor receptor (EGFR). The study finds a positive correlation between AREG expression and E2 treatment. Furthermore, inhibition of the AREG/EGFR signaling pathway using lapatinib in ER-positive breast cancer cells completely abrogated E2-induced cell proliferation and S-phase entry. Similarly, AREG knockdown resulted in decreased cell proliferation, S-phase induction, cell migration, and tumor growth, mirroring the effects observed with lapatinib treatment. The authors propose that AREG serves as an intermediary between EGFR and ER, and suggest that combination therapy targeting both ERs and EGFRs may effectively prevent tumor progression in patients with EGFR-positive/ER-positive breast cancer. The findings presented are significant and merit publication in Cells.
Specific comments
Pg 2 – lines 74 – 80. Please, remove cell lines not used in the article
Pg 3 – line 95, 2 x 103 (superscript)
Pg 7- Figure 2E – the violet line is not visible, correct.
Pg 10 - line 373 correct fig 6C to 6D
Pg 13 – line 413 – authors????
Author Response
Specific comments
Pg 2 – lines 74 – 80. Please, remove cell lines not used in the article
Thank you for your comment. The cell lines listed are utilized either in the main manuscript or in the supplementary data, and thus have been appropriately included.
Pg 3 – line 95, 2 x 103 (superscript)
Thank you for your comment. We corrected manuscript.
Pg 7- Figure 2E – the violet line is not visible, correct.
Thank you for your comment. We have revised Figure 2E by adding a text label to clearly identify the violet line, thereby improving its visibility.
Pg 10 - line 373 correct fig 6C to 6D
Thank you for your comment. We have corrected the figure citation in the main text, changing it from 6C to 6D.
Pg 13 – line 413 – authors????
Thank you for your comment. It’s my mistake. I’m so sorry. I corrected.

Round 2
Reviewer 1 Report
Comments and Suggestions for Authors
As mentioned in point 1, author should include discussion on previous literature on E2-boosting cell fitness in the text describing Figure 1, and E2-responsive AREG expression in ER+ breast cancer in the text describing Figure 2.
Authors should consider moving Figure S1C-E to main Figure 2. It makes it more compelling.
Author Response
- As mentioned in point 1, author should include discussion on previous literature on E2-boosting cell fitness in the text describing Figure 1, and E2-responsive AREG expression in ER+ breast cancer in the text describing Figure 2.
Thank you for your comment. According to your comment, we added sentence as follow “These effects were verified by confirming the effect of E2 on the cell proliferation, growth, and tumorigenicity of ER+ breast cancer cells [8]. (PMID: 37835383)”
In addition, we inserted as follow “However, AREG acts as a significant downstream effector of ER signaling and is directly involved in cell proliferation [19]. (PMID: 26527289).”
- Authors should consider moving Figure S1C-E to main Figure 2. It makes it more compelling.
Thank you. I totally agree with your comment. So, we transferred Figure S1C-E to Figure 2G–I and slightly corrected the results section and figure legends. “In addition, AREG knockdown cell lines did not significantly affect cell proliferation (Fig. 2G) and cell cycle progression (Fig. 2H and I) in response to E2 treatment.“
